# Work-Related, Non-Specific Low Back Pain among Physiotherapists in France: Prevalence and Biomechanical and Psychosocial Risk Factors, as a Function of Practice Pattern

**DOI:** 10.3390/ijerph20054343

**Published:** 2023-02-28

**Authors:** Baptiste Pellissier, François-Régis Sarhan, Frédéric Telliez

**Affiliations:** 1Institut de Formation en Masso-Kinésithérapie, CHU Amiens-Picardie, 30 Place Pr. Christian Cabrol, CEDEX 1, 80054 Amiens, France; 2Institut d’Ingénierie de la Santé-UFR de Médecine, Université de Picardie Jules Verne, 3 Rue des Louvels, 80036 Amiens, France; 3Equipe Chirurgie et Extrémité Céphalique Caractérisation Morphologique et Fonctionnelle UR 7516, Université de Picardie Jules Verne, CHU-Amiens, Place Pr. Christian Cabrol, CEDEX 1, 80054 Amiens, France; 4Laboratoire PériTox UMR_I 01, Centre Universitaire de Recherche en Santé, Université de Picardie Jules Verne, Chemin du Thil, 80025 Amiens, France

**Keywords:** physiotherapists, low back pain, practices, occupational risk factors, musculoskeletal disorders

## Abstract

Background. Physiotherapists worldwide experience lower back pain (LBP). Up to 80% of physiotherapists report having experienced an episode of LBP at some point in their career, and LBP is the most common musculoskeletal disorder in this profession. In France, the prevalence of LBP among physiotherapists and associated work-related risk factors have not previously been studied. Objective. To determine whether the risk of work-related non-specific LBP among French physiotherapists depends on practice pattern. Method. A link to an online self-questionnaire was sent to French physiotherapists. The various practice patterns were compared with regard to the prevalence of LBP, the total number of days with LBP during the previous 12 months, and the degree of exposure to biomechanical, psychosocial and organisational risk factors. Results. Among the 604 physiotherapists included in the study, the prevalence of work-related, non-specific LBP in the previous 12 months was 40.4%. The prevalence was significantly greater among physiotherapists working in geriatrics (*p* = 0.033) and significantly lower in sports medicine (*p* = 0.010). Differences in exposure to risk factors were also found. Conclusions. The risk of non-specific LBP among French physiotherapists appears to depend on the mode of practice. All the various dimensions of risk must be taken into account. The present study could serve as a basis for more targeted research on the most exposed practices.

## 1. Introduction

Lower back pain (LBP) is a major public health problem worldwide [1]. It particularly affects people of working age [2] and is the most common healthcare problem among workers in European countries [3]. Healthcare professionals are not spared: nurses, nurse assistants, dentists, paramedics, occupational therapists and physiotherapists can experience LBP [4,5,6,7,8,9,10,11,12].

According to two systematic reviews [13,14], up to 80% of physiotherapists report having experienced at least one episode of LBP during their career, and 73% at least one episode in the previous 12 months. LBP is the most frequent musculoskeletal disorder (MSD) among physiotherapists, ahead of neck/thorax, shoulder, wrist/hand and thumb problems.

Lower back pain is a multifactorial condition [1], and occupational factors reportedly account for 37% of the risk [15]. Physiotherapy-related biomechanical factors have been relatively well characterized. The main risks for physiotherapists are linked to major physical efforts (such as transfers and patient handling manoeuvres [12,16,17,18,19,20,21,22,23,24,25,26,27,28,29]), uncomfortable or prolonged working positions [5,18,19,20,21,22,23,24,25,26,27], trunk flexion and rotation movements [18,19,20,21,22,23,24,25,26,28] and reactions to a fall or a unexpected movement by the patient [12,17,18,19,20,21,22,23,24,25,26,27,28]. In terms of personal factors, recently qualified physiotherapists and female physiotherapists appear to experience LBP more [13,14]. In contrast, the psychosocial and organisational factors associated with work-related LBP among physiotherapists have rarely been studied. The results of a study by Campo et al. (2008) suggest that stress at work is a risk factor and that the psychosocial dimension has a major role in the development and persistence of MSDs [30].

According to several descriptive studies, the highest prevalences of LBP among physiotherapists are found in hospital settings [12,16,19,28,31], retirement homes [17] and rehabilitation centres [16,22,26,28,31]. Several clinical specialties have been considered (orthopaedics, neurology, paediatrics and geriatrics), with various LBP prevalence rates [16,17,19,24,27,31]. The physiotherapist’s type and field of practice thus appear to influence the risk of LBP. Nevertheless, a statistically significant relationship between MSDs, the practice setting [21,31] and/or the specialty [21] has never been reported.

As the leading occupational health problem among physiotherapists, LBP and its associated occupational risk factors are important issues both for the practitioners’ quality of life and the quality and safety of patient care. Indeed, providing optimal patient care is problematic if the physiotherapist is experiencing back pain; in a study conducted by West and Gardner (2001) in the USA, 92% of the participating physiotherapists stated that they had changed their techniques as a result of LBP [20]. Some used electrotherapy [21], and others decreased their amount of time in contact with the patient, changed or reduced the number of procedures or even changed their field or type of practice [18,20,21,24,26,27].

The objectives of the present study were thus to (i) determine whether the prevalence of LBP among physiotherapists is influenced by practice pattern and (ii) identify the biomechanical, psychosocial and organisational risk factors for non-specific LBP among physiotherapists as a function of their practices.

## 2. Materials and Methods

### 2.1. Design

We performed a retrospective, cross-sectional study with online recruitment and participation. A link to an online self-questionnaire (Appendix A) was sent to physiotherapists in France via social networks and the French National Council of Physiotherapists’ web site. Replies were collected between 20 November 2019 and 7 February 2020 (one month before the start of France’s first period of lockdown during the coronavirus disease 2019 epidemic).

### 2.2. Ethical Considerations

In line with French legislation, approval by an independent ethics committee was not required (simplified procedure, ASAP law (2020) amending art. L 1123-7 Public Health Code). This study was nevertheless performed in accordance with the ethical standards of the 1964 Declaration of Helsinki and its subsequent revisions. All data were stored securely, in line with the European Union’s General Data Protection Regulation and the guidelines issued by the French National Data Protection Commission (Paris, France) and registered under number 2222623. The questionnaire data were collected anonymously. Before filling out the questionnaire, all the participants provided their written consent. By giving their consent, participants confirmed that they understood (i) the study information, (ii) that data collected for research purposes would remain confidential, and (iii) that they could contact the research team if they had any further questions. 

### 2.3. Participants

We included physiotherapists practicing in France and who had treated patients during the previous 12 months. Physiotherapists were excluded if they had changed their type of practice or had qualified during the previous 12 months, if they worked for less than 30 h per week or if they had another job that accounted for more than 10% of their working time. Lastly, questionnaires with uninterpretable answers were excluded.

### 2.4. The Study Questionnaire

The study questionnaire was based on previously published surveys [20,21,23,24,25] and was adapted for use with French physiotherapists. The questionnaire comprised four sections (see Appendix A). The first section enabled us to select physiotherapists who met the inclusion criteria and to collect data on their age, sex and the following practice variables: employment status (a private practitioner or a salaried employee), practice setting (a private office and/or home care, a hospital setting or a rehabilitation centre), the type of disorders primarily treated (MSDs, neuromuscular disorders or respiratory, cardiovascular, internal organ or integumental disorders) and the clinical specialty (paediatrics, geriatrics or sports medicine), as defined in the French national classification of [32]). Henceforth, we shall use the term “practice pattern” to refer to employment status, practice setting, disorders primarily treated and clinical specialty. 

The second section of the questionnaire focused on the LBP ([1,21,33]). If the respondent had experienced LBP in the previous 12 months, he/she had to specify the total number of days with pain, whether a specific cause had been diagnosed, whether the LBP was primarily related to his/her professional activity, etc.

The third and fourth sections contained questions on the participants’ perceived working conditions. On a numerical scale ranging from 0 (never/not at all/positive perception) to 10 (always/extremely/negative perception), the participants had to rate their occupational exposure to the biomechanical risk factors mentioned in the literature and to psychosocial/organisational risk factors in the workplace. 

In order to assess demanding work tasks, a low degree of job control (usually defined as job strain) and poor social support (which are predictors of LBP) [34]), our questions were based on the Job Content Questionnaire [35]. We also added questions on dissatisfaction and hostility, as recommended more recently by Buruck et al. (2019) [34] in his Areas of Worklife model. These psychosocial occupational risk factors are also used in the “blue flags” guidelines on non-specific LBP [33].

### 2.5. Study Endpoints

The primary study endpoint was the prevalence of work-related, non-specific LBP in the previous 12 months. Only this type of LBP was included. Physiotherapists with specific LBP were identified through question 2.3, and their replies were excluded from our analysis. The secondary endpoints were the number of days with LBP, demographic characteristics (age and sex) and exposure to biomechanical and psychosocial/organisational risk factors (rated from 0 to 10). These data were compared as a function of four different practice variables: the employment status, the practice setting, the type of disorders primarily treated, and the clinical specialty. Data from physiotherapists with several concomitant types or fields of practice and data from subgroups smaller than n = 5 were not included in the comparisons. The sexes were also compared with regard to the prevalence of work-related, non-specific LBP.

### 2.6. Statistical Analysis

Data were processed using XLSTAT^®^ software (version 2020.1.1; Addinsoft, Paris, France) and JASP software (version 0.11.1.0; GNU Affero General Public License). The prevalences and the sex distributions were compared in a chi-squared test. If a statistically significant difference was detected, Fisher’s exact test was used to compare the observed and expected values in each group. For quantitative variables (e.g., age, number of days or exposure to risk factors), the normality of distribution was checked with the Shapiro–Wilk test. In fact, none of the variables in any of the groups were normally distributed; we therefore applied Kruskal–Wallis and Mann–Whitney tests. If a statistically significant difference was detected, a pairwise post-hoc test with correction for multiple comparisons was applied. The threshold for statistical significance was set to *p* < 0.05.

## 3. Results

In all, 720 replies were received (Figure 1). Thirteen replies were not included because the respondents had not treated any patients in the previous 12 months (n = 12) or were not practicing in France (n = 1). Of the 707 questionnaires included, 103 met one or more of the exclusion criteria and were not analysed; hence, 604 questionnaires were included in the final analysis.

In 2020, there were 90,315 physiotherapists in France [36]. With a sample size of 604, the results are considered to be accurate to ±2.92 percentage points (95% confidence interval) [37]. The study sample therefore comprised 604 physiotherapists (417 (69.0%) women and 187 (31.0%) men). The mean ± standard deviation (SD) age was 36.4 ± 10.1, and the average seniority was 13.1 ± 10.0 years. With regard to employment status, there were 491 (81.3%) private practitioners, 98 (16.2%) salaried practitioners and 15 (2.5%) practitioners with both private-practice and salaried activities (Table 1). 

### 3.1. Prevalence of Work-Related, Non-Specific LBP

The prevalence of LBP (of any type, whether work-related or not) was 81.0% for the career to date and 57.1% for the previous 12 months. The prevalence of work-related, non-specific LBP in the previous 12 months was 40.4%. The prevalence did not differ significantly by sex (37.4% among men and 41.7% among women; *p* = 0.320). Likewise, there were no significant differences with regard to the employment status, the type of practice, and the main disorders treated (Table 2). In contrast, clinical specialty was significantly associated with the prevalence of work-related, non-specific LBP (*p* = 0.007), which was greater in geriatrics (*p* = 0.033) and lower in sports medicine (*p* = 0.010).

### 3.2. Number of Days with LBP

The physiotherapists with work-related, non-specific LBP had experienced the condition for a median [interquartile range (IQR)] of 21 [10–42.75] days in the previous 12 months. The median number of days with non-specific LBP was significantly greater among private-practice physiotherapists than among salaried physiotherapists (22.5 [10–52.5] vs. 15 [7,8,9,10,11,12,13,14,15,16,17,18,19,20,21,22,23,24,25,26,27,28,29,30], respectively; *p* = 0.016). There were no significant differences for practice setting, types of disorders primarily treated, or clinical specialty.

### 3.3. Personal Factors

The proportion of female physiotherapists was significantly higher among those who primarily treated neuromuscular disorders (82.1%; *p* = 0.032) and was significantly lower among those treating MSDs (63.9%; *p* = 0.002) and those working in sports medicine (37.7%; *p* < 0.0001).

Physiotherapists who primarily treated MSDs (mean ± SD age: 35.3 ± 9.7; *p* < 0.001) or neuromuscular disorders (mean ± SD age: 34.3 ± 9.6; *p* = 0.007) were significantly younger than physiotherapists who did not specialize in a particular set of disorders (mean ± SD age =39.2 ± 10.7).

### 3.4. Biomechanical Risk Factors

The physiotherapists’ levels of perception of biomechanical risk factors by the type of practice are summarized in Table 3 and Appendix B. Salaried physiotherapists were significantly more exposed than private-practice physiotherapists to manual transfers of dependent patients (*p* < 0.001), lifting heavy loads (*p* < 0.001), working in an uncomfortable position (*p* = 0.004), and having to react to a fall or a sudden, unexpected movement by the patient (*p* < 0.001) (Table 3).

Physiotherapists involved in home care and those working in a hospital setting or a rehabilitation centre were significantly more exposed to these biomechanical factors than those working in a private office (*p* < 0.001 for all, except working in an uncomfortable position which was not different between a rehabilitation centre and a private office). The physiotherapists working in a private office were more exposed if they were involved in home care (*p* < 0.001 for manual transfers, lifting heavy loads and reacting to an unexpected movement), and the physiotherapists involved in home care were more exposed to working in an uncomfortable position than those working in a rehabilitation centre (*p* = 0.015).

Physiotherapists who primarily treated neuromuscular disorders were significantly more exposed to the four biomechanical factors than those treating MSDs (*p* < 0.001 for manual transfers, lifting heavy loads and reacting to an unexpected movement; *p* = 0.012 for working in an uncomfortable position). Exposure to manual transfers of dependant patients (*p* < 0.001) was significant less prevalent in physiotherapists who primarily treated MSDs relative to physiotherapists who treated (or were working in) no particular disorder.

With regard to clinical specialty, physiotherapists specializing in sports medicine were less exposed than those working in geriatrics (*p* < 0.001 for manual transfers and working in an uncomfortable position; *p* = 0.003 for lifting heavy loads and reacting to an unexpected movement) or in paediatrics (*p* < 0.001 for manual transfers and working in an uncomfortable position; *p* = 0.009 for reacting to an unexpected movement). Physiotherapists working in sports medicine were also less exposed to reactions to a fall or an unexpected movement by the patient (*p* = 0.006).

Compared with the lack of a particular clinical specialty, working in geriatrics (*p* < 0.001 for all four factors) or paediatrics (*p* < 0.001 for manual transfers; *p* = 0.011 for working in an uncomfortable position; *p* = 0.005 for reacting to an unexpected movement) were associated with significantly greater exposure to these biomechanical factors.

### 3.5. Psychosocial and Organisational Risk Factors

Private-practice physiotherapists worked significantly more hours per week than salaried physiotherapists did (mean ± SD: 46.6 ± 7.9 vs. 37.7 ± 3.4 h, respectively, *p* < 0.001) (Appendix C). Relative to the private-practice physiotherapists, the salaried physiotherapists estimated that their work environment was significantly more hostile (*p* < 0.001) and that they received less social support at work (*p* = 0.038). However, the private-practice physiotherapists felt more time pressure (*p* = 0.013) than the salaried physiotherapists (Table 4). 

With regard to dissatisfaction at work, a significant effect of the type of practice was present, but pairwise post-hoc tests failed to detect any significant intergroup differences. Physiotherapists involved in home care (*p* = 0.026) and those working in a hospital setting (*p* < 0.001) considered that their work environment was significantly more hostile than physiotherapists working in a private office. Compared with physiotherapists who did not specialize in treating a particular type of disorder, physiotherapists treating MSDs considered that they were less able to change their work procedures (*p* = 0.005) and that they received less social support at work (*p* = 0.018). With regard to the hostility of the work environment and time pressure, we observed a significant effect of clinical specialty but again failed to detect pairwise differences in post-hoc tests. Physiotherapists working in geriatrics perceived their work environment to be more hostile and considered that they were significantly more stressed at work than physiotherapists without a clinical specialty (*p* = 0.026 and *p* = 0.044) and those working in sports medicine (*p* = 0.005 and *p* = 0.026) did.

## 4. Discussion

In our study, the self-reported whole-career prevalence of LBP of any type among physiotherapists in France (81.0%) was higher than any of the literature values from studies conducted in other countries (26.0% to 79.6%). However, this was not the case for the prevalence of LBP in the previous 12 months (57.1% in France versus 22.0% to 73.1% for studies performed in other countries) [13,14]. The prevalence in the previous 12 months reported in the present study was lower than that described for other healthcare workers: 80% and 88.5% among nurses in studies by Jradi et al. (2020) [38] and Bryndal et al. (2022), respectively [39], 74% among operating room personnel [40], and 65% among obstetric care providers [41].

In our study, 244 (40.4%) of the 604 physiotherapists had experienced work-related, non-specific LBP at some time during the previous 12 months. The prevalence appears to be influenced by the type of physiotherapy activity in general and the clinical specialty in particular.

As also reported by Alrowayeh et al. (2010) for a study in Kuwait [21], we did not observe any association with employment status, practice setting, or the type of disorders primarily treated. However, the significant prevalence of work-related, non-specific LBP among physiotherapists involved in home care differs from the findings of Vieira et al. (2016) in the USA [17], where this mode of practice was associated with the lowest prevalence of LBP [17]. The prevalence of work-related, non-specific LBP among hospital-based physiotherapists in the present survey was also greater than the values reported in several descriptive studies conducted in other countries [12,16,19,28,31]. The greater exposure to manual transfers of dependant patients and the lifting of heavy loads in home care, hospital settings and geriatrics is in line with the results of Darragh et al.’s (2012) study in the USA [12]; according to the researchers, most of the physiotherapists attributed their LBP to patient transfers and handling.

Interestingly, we found that an episode of LBP lasted for longer in private-practice physiotherapists than in salaried physiotherapists. The greater perceived time pressure and longer working hours in private practice might explain this finding.

In contrast to the reports by Alrowayeh et al. (2010) in Kuwait and Cromie et al. (2000) in the USA, we evidenced a significant association between clinical specialty and the prevalence of LBP [21,31]. Geriatrics was the most affected specialty (prevalence: 52.8%), and sports medicine was the least affected (24.6%). The significantly greater prevalence of LBP observed in physiotherapists working in geriatrics is in line with Vieira et al.’s (2016) descriptive study in the USA [17], in which the prevalences of LBP in the previous 12 months were 71% for geriatric units and 100% for retirement homes. There are several possible explanations for these observations. The high proportion of men in sports medicine might be relevant because female sex appears to be a risk factor for LBP among physiotherapists [13,14]. Physiotherapists specializing in geriatrics were more exposed to biomechanical risks factors (such as manual transfers of dependant patients and lifting heavy loads). Salaried employment status, a hospital setting, a home care setting, the treatment of neuromuscular disorders, geriatrics and paediatrics were all associated with greater exposure to biomechanical risk factors in general and manual transfers of dependant patients and working in an uncomfortable position in particular. The psychosocial dimension might also have had a role because salaried physiotherapists and physiotherapists working in geriatric units, hospital settings or home care considered that they were more exposed to psychosocial constraints in general and a hostile work environment in particular. Thus, working in geriatrics might expose physiotherapists to a greater risk of LBP. Greater levels of dependence among elderly patients might require more physical effort and uncomfortable working positions and thus contribute to a greater risk for the lower back. 

The present study provided a large amount of new information on the risk of LBP as a function of the mode of physiotherapy practice. This is the first study to have investigated this topic in France. The sample was relatively large (n = 604) and the proportions of private-practice and salaried physiotherapists were representative of practitioners in France as a whole [36]. The proportions of the various types of practice and clinical specialities also matched our expectations. In contrast, the online recruitment method (i.e., the non-randomized inclusion of voluntary participants) was probably subject to selection bias and thus limited the sample’s representativeness. Technologies like social networks tend to attract a younger and female-biased population (most French physiotherapists under the age of 40 are women [37]). Thus, the percentage of women in our sample (69.0%) was not representative of physiotherapists in France as a whole (50.6% in 2020 [36]). Consequently, the prevalence levels observed here were perhaps overestimated. Moreover, the study’s retrospective design with the use of a self-questionnaire might have generated information bias (e.g., memory bias). Other study limitations included our pairwise comparisons of groups of physiotherapists with sometimes very different sample sizes.

The study questionnaire focused on work-related risk factors for LBP mentioned in the literature. In contrast, we did not take account of possible links between types of practice, even though the distribution of our sample of physiotherapists reflected their actual activity. However, it would have been difficult to categorize them more precisely. Our present results gave us an overview of occupational risks among physiotherapists in France; however, the results for each particular type or field of practice must be interpreted with caution. It would be also interesting to consider physiotherapists’ beliefs and attitudes, since these might influence their perceptions of occupational risk factors.

The present study constitutes a first step towards screening for at-risk occupational situations prior to an intervention in the field (e.g., a human factors analysis of care and patient management activities). Our results highlighted (i) the influence of clinical speciality on the prevalence of non-specific LBP among physiotherapists and (ii) some dominant risk factors (and thus targets for prevention) as a function of mode of practice. These results should help to raise physiotherapists’ awareness of their exposure to risk factors. Nevertheless, in order to set up optimal prevention actions, this work will have to be pursued. Our present results could serve as a basis for future in-depth research on the risk of LBP among physiotherapists. Although LBP is multifactorial, particular attention should be paid to certain aspects depending on employment status—notably, biomechanical risk factors (for salaried physiotherapists and physiotherapists involved in home care, the treatment of neuromuscular disorders, geriatrics and paediatrics) and organisational risk factors (for private-practice physiotherapists). The psychosocial dimension also warrants further investigation. Each practice pattern should be studied in more detail, and it will be necessary to consider the range of activities performed within a given specialty (the techniques used, the patients seen, etc.). In recent years, a large body of scientific data has led to the identification of effective treatments for LBP (such as Pilates and other exercises [42,43,44]), which can also be applied by physiotherapists. As mentioned by Modhi et al. (2022) [45], it would be also interesting to evaluate both the preventive measures and effective treatments applied by physiotherapists [36] and these measures’ impact on the quality of care.

## 5. Conclusions

The risk of LBP in physiotherapists appears to depend on the mode of practice in general and the clinical specialty in particular. Understanding these disparities will require further in-depth investigations. Thanks to its general approach, this study constitutes a first step towards characterizing risk factors for LBP among physiotherapists and could be used as a basis for more targeted research, such as a human factors analysis of risk factors, opportunities for prevention, and ways of reducing the risk of work-related, non-specific LBP among physiotherapists.

## Figures and Tables

**Figure 1 ijerph-20-04343-f001:**
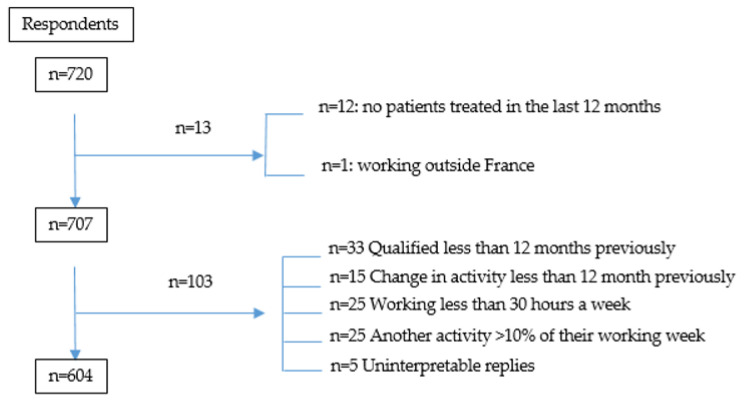
Study flow chart, describing the inclusion and exclusion of respondents.

**Table 1 ijerph-20-04343-t001:** Distribution of the participating physiotherapists, as a function of practice pattern.

Practice Pattern		Number	Percentage of the Whole Sample (%)	Number Considered in the Comparisons
Employment status	Private practitioner	491	83.3	n = 589
	Salaried employee	98	16.7	
Practice setting	Private office and home care	345	57.1	n = 551
	Private office	105	17.4	
	Home care	15	2.5	
	Rehabilitation centre	35	5.8	
	Hospital setting	51	8.4	
Disorders primarily treated	No particular disorders	186	30.8	n = 604
	Musculoskeletal disorders	335	55.5	
	Neuromuscular disorders	56	9.3	
	Respiratory/cardiovascular/internal organ/integumental	27	4.5	
Clinical specialty	No speciality	430	71.2	n = 598
	Geriatrics	72	11.9	
	Sports medicine	61	10.1	
	Paediatrics	35	5.8	

**Table 2 ijerph-20-04343-t002:** Prevalences of work-related, non-specific lower back pain, as a function of practice pattern.

Practice Pattern		Prevalence (%)	95% Confidence Interval	*p*-Value ^1^
Employment status	Private practitioner	41.6	37.2–45.9	*p* = 0.748
	Salaried employee	39.8	30.1–49.5	
Practice setting	Private office and home care	41.7	36.5–46.9	*p* = 0.106
	Private office	34.3	25.2–43.4	
	Home care	66.7	42.8–90.5	
	Rehabilitation centre	37.1	21.1–53.2	
	Hospital setting	49.0	35.3–62.7	
Disorders primarily treated	No particular disorders	42.5	35.4–49.6	*p* = 0.760
	Musculoskeletal disorders	39.4	34.2–44.6	
	Neuromuscular disorders	42.9	29.9–55.8	
	Respiratory/cardiovascular/internal organ/integumental	33.3	15.6–51.1	
Clinical specialty	No speciality	41.4	36.7–46.1	*p* = 0.007
	Geriatrics	52.8	41.2–64.3	
	Sports medicine	24.6	13.8–35.4	
	Paediatrics	31.4	16.0–46.8	

^1^ in a chi-squared test.

**Table 3 ijerph-20-04343-t003:** Results of statistical tests for exposure to biomechanical risk factors, as a function of practice pattern.

Practice Pattern	*p*-Value						
	High Physical Work Load	Manual Transfers of Dependant Patients	Lifting Heavy Loads	Working in an Uncomfortable Position	Trunk Flexion and Rotation Movements	Prolonged Work in the Same Position	Reacting to a Fall or a Sudden, Unexpected Movement by the Patient
Employment status ^1^ (n = 589)	NS	*p* < 0.001	*p* < 0.001	*p* = 0.004	NS	NS	*p* < 0.001
Practice setting ^2^ (n = 551)	NS	*p* < 0.001	*p* < 0.001	*p* < 0.001	NS	NS	*p* < 0.001
Disorders primarily treated ^2^ (n = 604)	NS	*p* < 0.001	*p* < 0.001	*p* < 0.001	NS	NS	*p* < 0.001
Clinical specialty ^2^ (n = 598)	NS	*p* < 0.001	*p* < 0.001	*p* < 0.001	NS	NS	*p* < 0.001

^1^: Mann–Whitney test. ^2^: Kruskal–Wallis test.

**Table 4 ijerph-20-04343-t004:** Results of statistical tests for exposure to psychosocial and organisational risk factors, as a function of practice pattern.

Practice Pattern	*p*-Value			
	Dissatisfaction at Work	Hostile Work Environment	High Demands at Work	Low Control over Work
Employment status (n = 589)	NS	*p* < 0.001	NS	NS
Practice setting ^2^ (n = 551)	*p* = 0.024	*p* < 0.001	NS	NS
Disorders primarily treated ^2^ (n = 604)	NS	*p* = 0.012	NS	NS
Clinical specialty ^2^ (n = 598)	NS	*p* = 0.006	NS	NS
Practice pattern	*p*-value			
	Lack of ability to change work practices	Lack of social support at work	Perceived time pressure	Stress at work
Employment status ^1^ (n = 589)	NS	*p* = 0.038	*p* = 0.013	NS
Practice setting ^2^ (n = 551)	NS	NS	NS	NS
Disorders primarily treated ^2^ (n = 604)	*p* = 0.008	*p* = 0.025	*p* = 0.027	NS
Clinical specialty ^2^ (n = 598)	NS	NS	NS	*p* = 0.016

^1^: Mann–Whitney test. ^2^: Kruskal–Wallis test.

## Data Availability

Suggested Data Availability Statements are available in section “MDPI Research Data Policies” at https://www.mdpi.com/ethics.

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
