# Peer review of "Work-Related, Non-Specific Low Back Pain among Physiotherapists in France: Prevalence and Biomechanical and Psychosocial Risk Factors, as a Function of Practice Pattern"

_ijerph, 2023, doi:10.3390/ijerph20054343_

Round 1

Reviewer 1 Report

  • Thank you for the opportunity to review this cross-sectional retrospective survey of French physiotherapists report of perceived work-related LBP and associated contributory factors. This is a large study with an appropriate number of participants from which to draw conclusions, albeit noting the limitations of self-report, recall and recruitment limitations. The paper details the survey used, and establishes that French physiotherapists perceive workplace mechanical demands, type of work, location of work, and both psychosocial and work organisational demands influence the prevalence and duration of low back pain symptoms in this population. The study strengths include the large sample size and the inclusion of “blue flags” or psychosocial and organisational factors.
  • While the survey development is well-described, it is unfortunate that the authors did not use standardised questionnaires, nor identify the beliefs of participants with respect to the causes of LBP. This is especially disappointing with respect to workplace psychosocial and organisational factors.
  • Providing the reader with some information on the numbers of physiotherapists in France, an estimate of sample size required for power calculations, and other demographics (all readily available but in French and from 2017), would have enabled insights into whether the survey and recruitment was representative.
    Recruitment relied on social media and participants to view the national physiotherapy organisational website, and as the authors indicate, the sample is therefore skewed towards younger people and women - and potentially those who have experienced LBP, also skewing the sample.
  • The authors suggest that physiotherapists present as a group with significant risk factors for LBP, but the global prevalence is similar to that reported in this study - some comparisons with both global prevalence and other occupational groups would be useful.
  • The most problematic aspect of this study is reliance on self-reported "work-relatedness", particularly without offering the reader some background on the medico-legal implications of this. If compensation, time off work, and treatment rely on LBP being identified as "work-related" this would skew the results.
  • Throughout the manuscript, the authors use the term "suffer" with respect to a person reporting an episode of LBP - this term implies a judgement about the impact of LBP, and it would be preferable to use a neutral term such as "experience".
  • The analysis is clear, uncomplicated, and while there are undoubtedly many other statistical analyses planned, as a descriptive study, these are adequate.
  • Line 262 is tautological - Our sample of 604 physiotherapists were particularly at risk of LBP because 40.4% had suffered from work-related, non-specific LBP during the previous 12 months. - being at risk of LBP is explained by having an episode of LBP. Comparing this prevalence with other epidemiological studies would have been more informative.
  • Overall, however, this study is useful and offers information from which other research can draw.

Author Response

Point by point reply – manuscriptijerph-2142384”

Work-Related, Non-Specific Low Back Pain among Physiotherapists in France: Prevalence and Biomechanical and Psychosocial Risk Factors, as a Function of the Practice Pattern.

Dear Editor and Reviewers,

Thank you very much for giving us your feedback and advice on how to improve our article. We have revised the manuscript accordingly. Please find below our point-by-point reply, which describes the revisions made in response to all the comments and questions. In the manuscript, the revisions are highlighted in blue.

  1. Review Report 1
  2. While the survey development is well-described, it is unfortunate that the authors did not use standardized questionnaires, nor identify the beliefs of participants with respect to the causes of LBP. This is especially disappointing with respect to workplace psychosocial and organizational factors.

Standardized questionnaire:

When preparing our study, we did not find any validated tools or data collection methods that were suitable for specifically studying a population of physiotherapists and their occupational (biomechanical and) psychosocial risk factors.

Part 1 of the questionnaire was designed to record the participants’ sociodemographic data: sex, age, and practice pattern (employment status, practice setting, disorders primarily treated, and clinical specialty).

Part 2 of the questionnaire focused on several aspects related to lower back pain.

As in the studies by Cromie et al. (2000) [21] and Bryndal et al. (2022) [39], musculoskeletal symptoms and/or low back pain were investigated using a self-administered, purpose-designed questionnaire. Like Cromie et al. and Bryndal et al., we also used a numerical (0-10) scale because it is (i) recommended by the French High Authority for Health (2022), and (ii) very well known, and widely used by healthcare professionals (including physiotherapists).

The biomechanical occupational risk factors for low back pain are often mentioned in the literature (Lundberg, 2015, Maher et al. 2017) (handling of heavy loads, awkward postures for the back, and whole-body vibrations), although some are still subject to debate (Balagué et al., 2012; Hartvigsen et al., 2018):

There are no specific questionnaires on LBP risk factors in physiotherapists. In our study, we wanted to target our questions by adapting and applying these biomechanical variables to the physiotherapists’ practice, and so we relied on the literature (refs. 5, 12, 16-29).

We also based our work on well-known literature tools for ergonomic assessments of biomechanical risk factors: the Rapid Upper Limb Assessment (RULA) (Mac Attamney and Corlett, 1993) and the Owako Working Posture Assessment System (OWAS) (Karhu et al., 1977, 1981) for back posture, and the French National Institute for Occupational Health and Safety’s specific guide (2017) on the manual handling of patients.

Part 4 of the questionnaire concerned psychosocial and organisational factors.

Demanding work tasks, a low degree of job control (usually defined as job strain) and poor social support (Karasek, 1979) are predictors of low back pain (Buruck et al., 2019; Lundberg 2015; Shaw et al., 2009; Loisel et Arena, 2013). These identified factors are well known psychosocial risk factors for low back pain (ref 34) and are considered in the “blue flags” guidelines (ref. 33).

To take account of your questions, we have added the following text:

Line 110-111: “The study questionnaire was based on previously published surveys [20,21,23-25] and was adapted for use with French physiotherapists.

Line 129-133: In order to assess demanding work tasks, a low degree of job control (usually defined as job strain), and poor social support (which are predictors of LBP) [34]), our questions were based on the Job Content Questionnaire [35]. We also added questions on dissatisfaction and hostility, as recommended more recently by Buruck et al. (2019) [34]. These psychosocial occupational risk factors are also used in the “blue flags” guidelines on non-specific LBP [33].”

Physiotherapists’ beliefs and attitudes are important and might influence patients’ beliefs and prognoses (Roussel et al., 2016; Gardner T et al. 2017; Hayden et al., Covchrane Data Base Syst. Rev. 2019). This is a very interesting question in the context of our study and a perspective to consider.

Line 326—328 We have add this suggestion to the Discussion It would be also interesting to consider physiotherapists’ beliefs and attitudes, since these might influence their perceptions of occupational risk factors.

  1. Providing the reader with some information on the numbers of physiotherapists in France, an estimate of sample size required for power calculations, and other demographics (all readily available but in French and from 2017), would have enabled insights into whether the survey and recruitment was representative.

We followed Gilliland and Melfi’s (2010) guidelines on calculation of the sample size and estimation of the confidence interval.

We added line 162-164: “In 2020, there were 90315 physiotherapists in France [36]. With a sample size of 604, the results are considered to be accurate to ±2.92 percentage points (95% confidence interval) [37].

Most of the studies of the prevalence of LBP and risk factors in occupational settings (see below) had smaller samples sizes than our study (except for Muaidi et al., 2016, and Holder et al., 1999). A sample of more than 200 respondents is often considered as being suitable in this kind of study.

Shehad et al. 2003: n=142 respondents

Vieira ER et al. 2016: n=121 respondents

Muaidi QI 2016: n=690 respondents

West DJ et al. 2001: n=217 respondents

Cromie et al. 2000: n=541 respondents

Adegoke BOA et al. 2008: n=128 respondents

Rozenfield et al. 2010: n= n = 182

Holder at al. 1999: n=667 respondents

Salik Y et al. 2004= n=120 respondents

Rugelj D 2003: n=113 respondents

Campo et al. 2008: n=590 respondents

Bryndal et al. 2022: n=544 physiotherapists and nurses

  1. Recruitment relied on social media and participants to view the national physiotherapy organizational website, and as the authors indicate, the sample is therefore skewed towards younger people and women - and potentially those who have experienced LBP, also skewing the sample.

We agree with this comment, and we discuss this point on lines 301-310 (in the first version, lines 310-319 in the revised version) and discuss the limitations of our method in the Discussion.

Online questionnaires and recruitment via social media offer other advantages, which we describe. In contrast, a mail-based survey requires to the respondent to mail back the reply, which often limits the response rate. In the studies by Adegoke et al. (2008) and Cromie et al., (2000), the response rates were respectively around 50% and 68%.

  1. The authors suggest that physiotherapists present as a group with significant risk factors for LBP, but the global prevalence is similar to that reported in this study - some comparisons with both global prevalence and other occupational groups would be useful.

Physiotherapists, nurses and other many professions are exposed to occupational risk factors for LBP. In the Discussion, the prevalence among physiotherapists is also discussed in the first paragraph: lines 258-261/ lines 263-266.

Lines 264-269 We have added “The prevalence in the previous 12 months reported in the present study was lower than that described for other healthcare workers: 80% and 88.5% among nurses in the studies by Jradi et al. (2020) [38] and Bryndal et al. (2022), respectively [39], 74% among operating room personnel [40], and 65% among obstetric care providers [41].

In our study, 244 (40.4%) of the 604 physiotherapists had experienced work-related, non-specific LBP at some time during the previous 12 months.

  1. The most problematic aspect of this study is reliance on self-reported "work-relatedness", particularly without offering the reader some background on the medico-legal implications of this. If compensation, time off work, and treatment rely on LBP being identified as "work-related" this would skew the results.

In our study, the replies were anonymous. It was clearly specified that the data were intended for research purposes only.The Reviewer's comment is interesting but would constitute a methodological problem if the physiotherapists’ responses were to be sent to an occupational disease compensation fund or health insurance fund. That was not the case here. 

  1. Throughout the manuscript, the authors use the term "suffer" with respect to a person reporting an episode of LBP - this term implies a judgement about the impact of LBP, and it would be preferable to use a neutral term such as "experience".

We have been advised by a native English-speaking PhD scientist that the idiomatic expression “to suffer from [a medical condition]” does not imply judgment or suffering (pain, distress, etc.) per se. For example, “I am suffering from a cold” simply means “I have a cold” to most native speakers – especially since a common cold is rarely painful or distressing. However, we have followed the reviewer’s suggestion and have replaced the term “suffer” by various other expressions, such as “experience”.

  1. The analysis is clear, uncomplicated, and while there are undoubtedly many other statistical analyses planned, as a descriptive study, these are adequate.

Thank you for this comment.

  1. Line 262 is tautological - Our sample of 604 physiotherapists were particularly at risk of LBP because 40.4% had suffered from work-related, non-specific LBP during the previous 12 months. Being at risk of LBP is explained by having an episode of LBP.

Lines 268-269: We have changed the sentence to “In our study, 244 (40.4%) of the 604 physiotherapists had experienced work-related, non-specific LBP at some time during the previous 12 months.”

Comparing this prevalence with other epidemiological studies would have been more informative.

This is covered in the first paragraphs of the Discussion, lines 260-277.

We sincerely hope that the revised manuscript meets the journal’s criteria for publication and would be delighted to provide any additional explanations, if required.

Thank you again for your interest in our work.

Yours sincerely,

  1. Pelissier, F.R. Sarhan, and F. Telliez.

Bibliography

Adegoke, B.O., Akodu, A.K. & Oyeyemi, A.L. Work-related musculoskeletal disorders among Nigerian Physiotherapists. BMC Musculoskelet Disord 9, 112 (2008). https://doi.org/10.1186/1471-2474-9-112

Balagué F, Mannion AF, Pellisé, Cedraschi C. Non-specific low back pain. Lancet 2012; 379:482-91

Chartered Society of Physiotherapy: Health and Safety Briefing Pack. No 11 Work- Related Strain Injuries (musculoskeletal disorders). 2001; CSP. London

Capodaglio EM. Comparison between the CR10 Borg’s scale and the VAS (visual analogue scale) during an arm-cranking exercise. J Occup Rehabil 2001 ;11(2):69–74.

Cromie JE, Robertson VJ, Best MO. Work- related musculoskeletal disorders in physical therapists: prevalence, severity, risks and responses. Phys Ther. 2000, 80: 336-351.

French National Institute for Occupational Health and Safety’s specific guide. Méthode d’analyse dela charge physique de travail-Secteur Sanitaire et Social. (2017; ED6291

Hartvigsen J, Hancock MJ, Kongsted A, Louw Q, Ferreira ML, Genevay S, et al. What low back pain is and why we need to pay attention. Lancet Lond Engl 2018; 391: 2356–67.

Karasek, R.A. Job Demands, Job Control, and Mental Strain: Implications for Job Redesign. Adm. Sci. Q. 1979, 24, 285–308

Karasek RA, Brisson C, Kawakami N, Houtman I, Bongers P, Amick B. The Job Content Questionnaire (JCQ): An Instrument for Internationally Comparative Assessments of Psychosocial Job Characteristics. Journal of Occupational Health Psychology 1998, Vol. 3, No. 4, 322-355.

Karhu O, Kansi P, Kuorinka I. Correcting working postures in industry: a practical method for analysis. 1977; Appl Ergon 8: 199-201.

Karhu O, Härkönen R, Sovali P, Vepsäläinen P. Observing working postures in industry: examples of OWAS application. 1981; Appl Ergon; 12: 13-17.

Kuorinka I, Jonsson B, Kilbom A, et al. Standardised Nordic questionnaires for the analysis of musculoskeletal symptoms. Applied Ergonomics. 1987;18:233–237.

Mierzejewski M, Kumar S: Prevalence of low back pain among physical therapists in Edmonton, Canada. Disabil Rehabil. 1997, 19 (8): 309-317.

Waongenngarm P. et al. Can the Borg CR‑10 scale for neck and low back discomfort predict future neck and low back pain among high‑risk office workers? International Archives of Occupational and Environmental Health (2022) 95:1881–1889.doi.org/10.1007/s00420-022-01883-3

Reviewer 2 Report

See attached

Author Response

Point by point reply – manuscriptijerph-2142384”

Work-Related, Non-Specific Low Back Pain among Physiotherapists in France: Prevalence and Biomechanical and Psychosocial Risk Factors, as a Function of the Practice Pattern.

Dear Editor and Reviewers,

Thank you very much for giving us your feedback and advice on how to improve our article. We have revised the manuscript accordingly. Please find below our point-by-point reply, which describes the revisions made in response to all the comments and questions. In the manuscript, the revisions are highlighted in blue.

  1. Review Report 2
  2. Abstract, line 23: “practice patterns” should be “practice pattern” I believe, here and throughout the rest of the manuscript.

We have replaced “practice patterns” with “practice pattern” throughout the manuscript.

  1. Page 2, line 91: “data was” should be “data were”.

Line 95: We have replaced “data was” with “data were”.

  1. Page 3, lines 131-132: The sentence, “Only this type of LBP was included because the physiotherapists judged that non-specific LBP was primarily related to their professional activity” require further explanations and/or a reference. I’m unclear how the authors determined this assertion? Perhaps if the survey respondents answer yes to question 2.3, they were eliminated from the data analysis?

Lines 137-138: We changed the text to “Physiotherapists with specific LBP were identified through question 2.3, and their replies were excluded from our analysis.”

This is also mentioned on lines 159-160 “Of the 707 questionnaires included, 103 met one or more of the exclusion criteria and were not analyzed; hence, 604 questionnaires were included in the final analysis.”

  1. Page 3, Section 2.4. The Study Questionnaire: did the authors create the questionnaire and if so, did they conduct psychometric testing on the questionnaire? Other than the reference to using “blue flags” guidelines, there is no information about the source of the questions in the survey. The survey is therefore a significant limitation. For example, questions 3.2 to 3.7 in the survey use the phrase “do you often….”. It does not appear that the word “often” has been defined and the meaning of the word “often” could wary among respondents and introduce variability in survey responses. How did the authors guard against this variability?

Unfortunately, we did not conduct psychometric testing on the questionnaire. We did not find any validated tools that were suitable for specifically studying a population of physiotherapists and their occupational (biomechanical and) psychosocial risk factors.

However, the study questionnaire was based on previous published questionnaires and was adapted for use with French physiotherapists (Cromie et al. 2000; Mierzejewski et al. 1997; Chartered Society of Physiotherapy, 2001; Lotters F et al., 2003, Palmer KT et al. 2007; Glover et al., 2005; West and Gardner al. 2001, Rugelj et al. 2003; Adegoke, BO et al. 2008). All these cited studies used self-questionnaires.

Part 1 of the questionnaire was designed to record the participants’ sociodemographic data: sex, age, and practice pattern (employment status, practice setting, disorders primarily treated, and clinical specialty).

Part 2 of the questionnaire focused on several aspects related to lower back pain.

As in the studies by Cromie et al. (2000) [21] and Bryndal et al. (2022), musculoskeletal symptoms and/or low back pain were investigated using a self-administered, purpose-designed questionnaire. Like Cromie et al. and Bryndal et al., we also used a numerical (0-10) scale because it is (i) recommended by the French High Authority for Health (2022), and (ii) very well known, and widely used by healthcare professionals (including physiotherapists).

The biomechanical occupational risk factors for low back pain are often mentioned in the literature (Lundberg, 2015, Maher et al. 2017) (handling of heavy loads, awkward postures for the back, and whole-body vibrations), although some are still subject to debate (Balagué et al., 2012; Hartvigsen et al., 2018):

There are no specific questionnaires on LBP risk factors in physiotherapists. In our study, we wanted to target our questions by adapting and applying these biomechanical variables to the physiotherapists’ practice, and so we relied on the literature (refs. 5, 12, 16-29).

We also based our work on well-known literature tools for ergonomic assessments of biomechanical risk factors: the Rapid Upper Limb Assessment (RULA) (Mac Attamney and Corlett, 1993) and the Owako Working Posture Assessment System (OWAS) (Karhu et al., 1977, 1981) for back posture, and the French National Institute for Occupational Health and Safety’s specific guide (2017) on the manual handling of patients.

Part 4 of the questionnaire concerned psychosocial and organisational factors.

Demanding work tasks, a low degree of job control (usually defined as job strain) and poor social support (Karasek, 1979) are predictors of low back pain (Buruck et al., 2019; Lundberg 2015; Shaw et al., 2009; Loisel et Arena, 2013). These identified factors are well known psychosocial risk factors for low back pain (ref 34) and are considered in the “blue flags” guidelines (ref. 33).

To take account of your questions, we have added the following text:

Line 110-111: “The study questionnaire was based on previously published surveys [20,21,23-25] and was adapted for use with French physiotherapists.

Line 129-133: In order to assess demanding work tasks, a low degree of job control (usually defined as job strain), and poor social support (which are predictors of LBP) [34]), our questions were based on the Job Content Questionnaire [35]. We also added questions on dissatisfaction and hostility, as recommended more recently by Buruck et al. (2019) [34] in his Areas of Worklife model. These psychosocial occupational risk factors are also used in the “blue flags” guidelines on non-specific LBP [33].”

The word often is a French misnomer/semantic form and was only used in part 3 of the questionnaire. Maybe, it would be not suitable to translate this term. The rate scale is clear with 0 Never to 10 Always and did not require to refer to the word “often”. The word often was not considered by the respondent.

We have deleted the word “often” in part 3.

  1. Page 8, line 246, “did” is not necessary at the end of the sentence

Line 250: “did” has been deleted.

  1. Page 9, line 301; “contributing” should be “contribute”.

Line 303 (revised version): We have replaced “contributing” with “contribute”.

  1. Page9, line 320: “taken account” should be “take account”

Line 322 (revised version): We have replaced “taken account” with “take account”.

  1. Page 10, lines 341-343: the authors mention the preventive measures physiotherapists may engage in as a potential factor to investigate in future research. I think this issue is a significant weakness of the present study. It is known that the most effective exercise for low back pain is Pilates, for example (Fernandez-Rodriguez, R. et al, Best exercise options for reducing pain and disability in adults with chronic low back pain: Pilates, strength, core-based and mind-body. A network meta-analysis. JOSPT In Press). How many of the respondents to the survey perform regular Pilates and what impact does that have on the frequency of LBP, etc.?

Thank you for your comment. Indeed, these treatment options can be recommended for LBP. In this sentence, we talked about preventing LBP for physiotherapist as an at-risk profession, i.e. avoiding a initial episode of related-work LBP.

To clarify, we have added the following sentences line 341: In recent years, a large body of scientific data has led to the identification of effective treatments for LBP (such as Pilates and other exercises [42,43,44]), which can also be applied by physiotherapists. As mentioned by Modhi et al. (2022) [45], it would be also interesting to evaluate both the preventive measures and effective treatments applied by physiotherapists [36] and these measures’ impact on the quality of care.

We sincerely hope that the revised manuscript meets the journal’s criteria for publication and would be delighted to provide any additional explanations, if required.

Thank you again for your interest in our work.

Yours sincerely,

  1. Pelissier, F.R. Sarhan, and F. Telliez.

Bibliography

Adegoke, B.O., Akodu, A.K. & Oyeyemi, A.L. Work-related musculoskeletal disorders among Nigerian Physiotherapists. BMC Musculoskelet Disord 9, 112 (2008). https://doi.org/10.1186/1471-2474-9-112

Balagué F, Mannion AF, Pellisé, Cedraschi C. Non-specific low back pain. Lancet 2012; 379:482-91

Chartered Society of Physiotherapy: Health and Safety Briefing Pack. No 11 Work- Related Strain Injuries (musculoskeletal disorders). 2001; CSP. London

Capodaglio EM. Comparison between the CR10 Borg’s scale and the VAS (visual analogue scale) during an arm-cranking exercise. J Occup Rehabil 2001 ;11(2):69–74.

Cromie JE, Robertson VJ, Best MO. Work- related musculoskeletal disorders in physical therapists: prevalence, severity, risks and responses. Phys Ther. 2000, 80: 336-351.

French National Institute for Occupational Health and Safety’s specific guide. Méthode d’analyse dela charge physique de travail-Secteur Sanitaire et Social. (2017; ED6291

Hartvigsen J, Hancock MJ, Kongsted A, Louw Q, Ferreira ML, Genevay S, et al. What low back pain is and why we need to pay attention. Lancet Lond Engl 2018; 391: 2356–67.

Karasek, R.A. Job Demands, Job Control, and Mental Strain: Implications for Job Redesign. Adm. Sci. Q. 1979, 24, 285–308

Karasek RA, Brisson C, Kawakami N, Houtman I, Bongers P, Amick B. The Job Content Questionnaire (JCQ): An Instrument for Internationally Comparative Assessments of Psychosocial Job Characteristics. Journal of Occupational Health Psychology 1998, Vol. 3, No. 4, 322-355.

Karhu O, Kansi P, Kuorinka I. Correcting working postures in industry: a practical method for analysis. 1977; Appl Ergon 8: 199-201.

Karhu O, Härkönen R, Sovali P, Vepsäläinen P. Observing working postures in industry: examples of OWAS application. 1981; Appl Ergon; 12: 13-17.

Kuorinka I, Jonsson B, Kilbom A, et al. Standardised Nordic questionnaires for the analysis of musculoskeletal symptoms. Applied Ergonomics. 1987;18:233–237.

Mierzejewski M, Kumar S: Prevalence of low back pain among physical therapists in Edmonton, Canada. Disabil Rehabil. 1997, 19 (8): 309-317.

Waongenngarm P. et al. Can the Borg CR‑10 scale for neck and low back discomfort predict future neck and low back pain among high‑risk office workers? International Archives of Occupational and Environmental Health (2022) 95:1881–1889.doi.org/10.1007/s00420-022-01883-3